# Observational Analyses of Dramatic Developments of A Severe Air Pollution Event in the Beijing Area

Ju Li[1], Jielun Sun[2], Mingyu Zhou[3], Zhigang Cheng[1], Qingchun Li[1], Xiaoyan Cao[1], and Jingjiang Zhang[1]

[1]Institute of Urban Meteorology, China Meteorological Administration, Beijing 100089, China
[2]National Center for Atmospheric Research, Boulder, Colorado, USA
[3]National Marine Environment Forecast Center, Beijing, China

*Correspondence to:* Jielun Sun (jsun@ucar.edu)

**Abstract.** A rapid development of a severe air pollution event in Beijing, China at the end of November 2015 was investigated with unprecedented observations collected during the field campaign of the Study of Urban Rainfall and Fog/Haze (SURF-15). Different from previous statistical analyses of air pollution events and their correlations with meteorological environmental conditions in the area, the role of turbulent mixing in the pollutant transfer was investigated in detail. The analyses indicate that the major pollution source associated with high particulate matter of diameter 2.5 $\mu$m ($PM_{2.5}$) was from south of Beijing. Before the dramatic $PM_{2.5}$ increasing day, the nighttime downslope flow from the mountains on the west-to-north side of Beijing reduced the surface $PM_{2.5}$ concentration northwest of Beijing. The nighttime surface stable boundary layer (SBL) not only kept the relatively less polluted air near the surface but also shielded the rough surface from the pollutant transfer by southwesterly wind above the SBL, leading to the fast transport of pollutants over the Beijing area at night. As the daytime convective turbulent mixing developed in the morning, turbulent mixing transported the elevated polluted air downward even though the weak surface wind was from northeast, leading to the dramatic increase of the surface $PM_{2.5}$ concentration in the urban area. As a result of both turbulent mixing and advection processes with possible aerosol growth from secondary aerosol formation under the low wind and high humidity conditions, the $PM_{2.5}$ concentration reached over 700 $\mu$g m$^{-3}$ in the Beijing area by the end of the day. Contributions of the two transporting processes to the $PM_{2.5}$ oscillations prior to this dramatic event were also analyzed. The study demonstrates the important role of large-eddy convective turbulent mixing in vertical transfer of pollutants and the role of the SBL in not only decoupling vertical transport of trace gases and aerosols but also in accelerating horizontal transfer of pollutants above.

## 1   Introduction

Rapid urbanization in China associated with its fast growing economy has led to heavy air pollution, which has drawn international attention. Although the number of "blue sky" days measured by the air quality index (AQI) increased in 2015 from its 2014 number according to the state bulletin issued by the Chinese Ministry of Environmental Protection (Chinese EPA), air pollution is still a serious issue especially in the region of Beijing-Tianjin-Hebei (BTH) and its surrounding area including provinces of Shanxi, Shandong, Henan, and Inner Mongolia. Heavy pollution events occurred frequently in the BTH region and its surrounding area. In 2015, 13 cities in the BTH region that are monitored by the Chinese EPA failed to meet the Chinese

air quality standard for 47.6% of the year with light, medium, heavy, and severe pollution days at 27.1%, 10.5%, 6.8% and 3.2% of the year, respectively.

Air pollution occurrence frequency, intensity, and impact areas are major concerns in China. An increasing number of studies have focused on Chinese air pollution and haze issues (e.g., Liang and Tang, 2017; Liu et al., 2017; Wang et al., 2017; Zhang et al., 2017a, b; Zhong et al., 2017; Wang et al., 2018). The worst pollution season in the Beijing area is during fall and winter between October and March although pollution may also occur frequently in summer (e.g., Zhao et al., 2012; Li et al., 2016). A number of studies have found that heavy pollution in Beijing is often associated with surface low pressures and weak winds (e.g., Meng and Cheng, 2002). Because Beijing is surrounded by the mountains from west to north, local circulations that are thermally and mechanically forced by the mountains, land-sea contrasts from Bohai east of Beijing, and urban canopies also contribute air pollution transportation in the Beijing area (e.g., Chen et al., 2009; Liu et al., 2009).

Zhang et al. (2006) indicated that the planetary boundary layer (PBL) depth is associated with the expansion volume for pollutants; low PBLs such as the stable boundary layer (SBL) provide favorite conditions for development of heavy pollution events. Xu et al. (2005), Zhao et al. (2013), and Liao et al. (2014) investigated correlations between environmental conditions of the atmospheric boundary layer and relatively high pollution events and also emphasized the role of the SBL in trapping pollutants. Wang et al. (2014a) reported that explosively developed and long-lasting air pollution events in Beijing are commonly associated with weak winds from weak pressure systems, the SBL, and high humidity. They also pointed out that rapid disappearance of air pollution is commonly due to strong northernly winds associated with a surface high pressure centered west of Beijing. Air pollution enhances concentration of particulate matter with diameter 2.5 $\mu$m (PM$_{2.5}$), which affect human health and reduce visibility.

Human activities contribute increased water vapor release in urban areas compared to rural areas as observed by Dou et al. (2014) especially in the nighttime SBL in winter. Pollutant contents have been investigated by Xu et al. (2005), Wang et al. (2014a), and Li et al. (2016). Han and Zhang (2014) found relative humidity about 10% to 40% higher than the annual average value was often connected with heavy pollution events in the Beijing area and its surrounding provinces. Zhang et al. (2012) investigated high-humidity enhancement of hygroscopic growth of aerosols. Zhang et al. (2007) analyzed chemical characteristics of seasonal variations of PM$_{2.5}$ in Beijing. Sun et al. (2014) found that high humidity could also enhance heterogeneous chemical reactions with highly concentrated mineral particles to increase generation of secondary aerosols. Quan et al. (2014) found that during heavy pollution events, significant amounts of NO$_3$ and SO$_4$ particles can be produced by gas-particle transformation from NO$_x$ and SO$_2$. Zhao et al. (2013) observed secondary formation of aerosols including inorganic and organic pollutants and emphasized heterogeneous reaction processes in contribution of sulfate and nitrate to the PM$_{2.5}$ concentration.

Understanding air pollution consists of two general aspects: aerosol transporting and generation mechanisms; two processes are highly correlated. Atmospheric circulations controlled by mechanical and thermal forcing provide conditions for not only aerosol transport but also environment for aerosol formation and growth. Although there is an increasing number of publications on air pollution transport in the Beijing area, most of them concentrated on statistical relationships among PM$_{2.5}$ concentrations, atmospheric variables, and weather patterns based on numerical models with limited observations. Although

there are modeling investigations on regional transport of pollutants (e.g., Wang et al., 2014b), few are on detailed observational analyses of pollutant transporting mechanisms due to lack of the spatial coverage of meteorological and aerosol observations especially turbulent mixing in pollutant transport in the Beijing area. In this study, we concentrate on a severe air pollution event through detailed analyses of aerosol transporting mechanism with both turbulence and aerosol observations from an

intensified field campaign in the Beijing area.

During the seven-days period between 26 November and 2 December 2015, an extremely heavy pollution event with the daily mean AQI greater than 200 developed in the Beijing area. The event would be classified as "red alert" by the Chinese EPA. However based on the local forecast, the pollution intensity of "orange alert" was issued instead. The failure of issuing the correct warning was publicized widely. As far as we know, three studies have investigated this event. Both Fan et al. (2016)

and Zhang et al. (2017a) studied the case with the WRF meoscale numerical model, and Hao et al. (2017) focused on the role of humidity in this event at Tianjing. Although Zhang et al. (2017a) concluded that the spatial pattern of the $PM_{2.5}$ concentration for this event could be predicted by the WRF-Chem model, the model failed to predict its peak value. During this event, the $PM_{2.5}$ concentration oscillated dramatically starting at 12 hours before the final rapid increase on 30 November. This event happened to occur during the three-year field campaign of the Study of Urban Rainfall and Fog/Haze (SURF-15) (Liang et al.,

2017), which started in the spring of 2015. Using extra measurements from SURF-15 as well as routine measurements across the Beijing area described in section 2, we investigate physical processes that led to the heavy pollution event in this study. We first study the regional environment related to the development of the heavy pollution event in section 3. We then examine physical transporting processes responsible for the dramatic oscillations of $PM_{2.5}$ on 29 November prior to the dramatic increase of $PM_{2.5}$, during the rapid development of the severe pollution event on 30 November, and the dramatic decrease of

the high $PM_{2.5}$ on the evening of 1 December in section 4. The summary is in section 5.

## 2   Instrumentation and Observations

The observations (http://www.ium.cn:8088/dataCenter/) were collected during SURF-15 sponsored by Institute of Urban Meteorology (IUM). We mainly focus on the data collected at the research site operated by the Institute of Atmospheric Physics (IAP), Chinese Academy of Sciences. All the observation heights used in this study are above the ground level (AGL). At

the IAP site, a 325-m research tower has been operational since 1978. Wind directions (020C, American MetOne) and speeds (010C, American MetOne), air temperature, and relative humidity (HC2-S3, Rotronic, Swiss) were measured at 15 levels (8, 15, 32, 47, 65, 80, 100, 120, 140, 160, 180, 200, 240, 280, 320 m) on the tower with the sampling rate of one sample per 20 s and averaged to one per min. Wind speed at each level was measured on two booms: one pointed to 315 deg from north and the other one to 135 deg; the larger value of the two is chosen as the wind speed for the level. Wind directions were measured

by only one sensor at each level. Three-dimensional sonic anemometers—each on a 2.2 m boom from the tower center—were installed at 47 m (Windmaster, Gill), 140 m (CSAT3, Campbell Scientific), and 280 m (CSAT3, Campbell Scientific) at the direction of 30°, 210°, and 230° relative to the center of the tower, respectively. Downward and upward pointing pyrgeometers and pyranometers (CNR1, Kipp & Zonen), and $CO_2$/ $H_2O$ concentrations sensors (LI-7500) were also installed at these three

levels. The data from the sonic anemometers and the LI-7500's were sampled at 10 samples per s. The turbulence data were calculated at every 30-min segment using the EDDY-PRO software (Burba, 2013). We also use a surface station about 20 m south of the IAP tower, where wind speed and direction (05103-L, R. M. Young), and temperature and relatively humidity (HMP45C, Vaisala) at 2.2 m were measured at one sample per 2 s.

In addition, a Doppler lidar (Windcube 100S, Leosphere) and a mini-micropulse aerosol lidar (mini-MPL, SigmaSpace) were located at about 20 m west-northwest of the IAP tower; the two lidars were 3-m apart. To observe wind profiles, the Doppler Beam Swing (DBS) mode was used at every 5-s interval with the vertical resolution of 20 m (Campbell et al., 2002), and the data were averaged to 1-min segments. Due to the optical overlap function issue (Hey et al., 2011), the lowest Doppler lidar observation height was set to 70 m based on comparison between lidar and in-situ observations even though its first range gate

of 50 m is issued by the manufacturer. The carrier-to-noise ratio (CNR), which is found to be highly correlated with aerosol backscatter (Aitken et al., 2012), as well as wind speeds and directions were observed from the Doppler lidar. Normalized relative backscatters (NRB) were derived from the mini MPL lidar measurement with the sampling rate of one sample per 30 s and the vertical resolution of 30 m above 100 m.

    Furthermore, we use remote sensed measurements from a mini-MPL at IUM, a ceilometer lidar (CL51) at Finnish Embassy

(FIN), a wind profiler (CFL-08) at Nanjiao Guanxiang Tai (NGT) and at Haidian (HAI) located 7 km to the northwest of the IAP tower (all these locations are marked in Fig. 1). We also use radio soundings launched twice a day at NGT, which is about 20 km southeast of the IAP tower. During the study period, daily MODIS (Moderate Resolution Imaging Spectroradiometer) Rapid Response satellite images over the Beijing Area are available from AERONET (https://aeronet.gsfc.nasa.gov/).

    We also use measurements from the dense network of automatic weather stations (AWS) operated by Beijing Meteorological

Service (BMS). These data include wind speed and direction (EL15-2C) at 10 m, temperature and humidity (HMP155, Chinese Huayun Company) at 1.5 m, and pressure (PTB210, Chinese Huayun Company) at 1.5 m observed at Changping (CHP), Chaoyang (CHA), Mentougou (MEN), Daxing (DAX), Tongzhou (TON), and HAI. In addition, we also use the hourly $PM_{2.5}$ concentration measured at $\sim 2$ m at these stations operated by BMS as part of their air quality monitoring network and the $PM_{2.5}$ concentration at $\sim 10$ m at the IAP site sampled at every minute. The $PM_{2.5}$ concentration was measured with sensors

from the LGH-01E, Landun Photoelectron Company at HAI, MEN, and TON, with Grimm 180 at CHA, CHP, and DAX, and with Thermo SHARP5030 at IAP.

## 3   Regional environment in Beijing and its surrounding area

On 26 November, the surface $PM_{2.5}$ concentration begun to increase steadily (Fig. 2). Starting from 0500 LST, 30 November, the $PM_{2.5}$ concentration increased dramatically and reached to 732 $\mu g\ m^{-3}$ at HAI and 786 $\mu g\ m^{-3}$ at MEN at 1500 LST on

1 December.

    Horizontally, a weak high pressure ridge at 850-mb moved from west to east of Beijing during 29-30 November (Fig. 3). Because of the movement of this pressure ridge, the wind at the upper PBL changed from northwesterly on 29 November to southwesterly on 30 November, which is consistent with the sounding observations at NGT (Fig. 4). The wind profiler

observations at both NGT and HAI indicate that the wind direction change occurred about midnight of 29 November (Fig. 4). On the surface, Beijing was between a high pressure on northwest and a low pressure on southeast on 29 November resulting in an easterly flow in Beijing. As the weak high pressure moved eastward, Beijing was between a weak high pressure on the east and a low pressure on the west on 30 November and was under influence of weak northeasterly winds (Fig. 3).

Vertically, the 2000-LST sounding at NGT on 29 November shows that there was a well-mixed layer above about 200 m up to about 1400 m indicating that the convective turbulent mixing during the daytime of 29 November was relatively strong (Fig. 4). The 0800-LST sounding on 30 November indicates that a well mixed convective boundary layer (CBL) started to develop near the surface with the overnight SBL above, which was being entrained into the CBL below. The relatively well-mixed residual layer from the previous daytime mixing was above the elevated SBL (Fig. 4). The CBL on 30 November was associated
with weak northeasterly winds, and the stable layer above was dominated by southwesterly winds. The relative humidity was highest in the stable layer compared to the residual layer above and the newly developed CBL below. Within the stable layer, the relative humidity decreased with height and dropped sharply at its bottom due to the turbulent mixing in the CBL below. By 2000 LST on 30 November, the surface layer changed to a SBL of near 100 m with the daytime well-mixed air above. The wind direction in the daytime convective boundary layer above the newly developed SBL became southernly and the bottom
SBL was northeasterly to easterly. In addition, the air humidity increased in both the SBL and the daytime convective layer above. The increased humidity is consistent with the reported visibility of less than 1 km at most AWS stations around Beijing on 30 November.

At 0100 LST, 30 November, the surface $PM_{2.5}$ concentration was high south of Beijing (Fig. 5). This polluted air was visibly confined east of the Taihang Mountain Ridge from the MODIS images on 29 and 30 November (Fig. 6), which is
often observed in winter (e.g., Xu et al., 2005; Zhao et al., 2013; Zhong et al., 2017). During the early morning hours of 30 November, for example, at 0500 LST, the surface maximum $PM_{2.5}$ concentration in Fig. 5 increased and its high concentration center was shifted toward southeast of Beijing (Fig. 5). From 0500 LST to 1100 LST, the $PM_{2.5}$ concentration in Beijing increased dramatically, and its development was faster than its surrounding area, which is likely due to the rough surface of the urban area in its contribution to stronger turbulent mixing (Sun et al., 2017). In the following section, we demonstrate that
the shift of the high surface $PM_{2.5}$ to southeast of Beijing results from the relatively clean flow downslope from the mountains northwest of Beijing, leading to the reduction of the surface $PM_{2.5}$ concentration northwest of Beijing. The northeastward movement of the high surface $PM_{2.5}$ air during the daytime is due to downward transfer of pollutants by convective mixing, where pollutants are supplied by the northeastward flow through horizontal advection above the newly developed CBL in the morning.

**4   Physical transporting processes in development of the heavy pollution event in the Beijing area**

In general, turbulence is generated either by positive buoyancy or by shear. Turbulent mixing leads to vertical transport of heat from the diurnal heating and cooling of the surface, resulting in air temperature changes. Turbulent mixing also leads to the vertical momentum transfer resulting in wind speed and direction changes because the background wind changes with height

especially near the surface due to surface drag. Turbulent mixing during daytime is commonly generated by convection from the warm air at the heated surface from solar heating. Once wind speeds are high, wind shear near the surface can also generate turbulence, which can occur day and night although shear is the dominant turbulence generation mechanism at night. At night, the air near the surface is cooled by molecular thermal conduction from the cold surface due to longwave emission. The vertical

temperature difference between the warm air heated from the daytime and the cooled air near the surface leads to the SBL, which reduces turbulence intensity and vertical turbulent transports of momentum, heat, trace gases and aerosols. As a result, the air-surface coupling and the surface emitted trace gases of water vapor and $CO_2$ are confined near the surface. Impacts of the surface roughness on air motions are also limited to the near-surface stable layer (e.g., Sun et al., 2016). In summary, the diurnal variation of solar radiation leads to the diurnal variation of air temperature, wind speed and direction, trace gases, and

aerosols near the surface.

During the first three days of our study period, 26 to 28 November, the maximum downward solar radiation decreased steadily (Fig. 7a). The low downward solar radiation on 28 November was consistent with the cloudy MODIS satellite image that day (Fig. 6a). The $PM_{2.5}$ concentration, air potential temperature $\theta$, specific humidity $q$, and $CO_2$ concentration increased steadily with their diurnal variations associated with turbulent mixing embedded in their increasing trends (Fig. 7). The wind

speed was high during the first part of 26 November, and was low for the rest of the three days in association with the regional low pressure system (Fig. 7g), which was consistent with the decreasing pressure observed at IAP (Fig. 7e). The wind direction within the IAP tower layer varied diurnally—westerly in the first half of each day and easterly during the second half, especially when the ambient wind speed was low, for example, on 27 and 28 November (Fig. 7f)—indicating the mountain west of Beijing played an important role in the diurnal wind-direction variation. Because of the contribution of the positive

buoyancy to turbulent mixing during daytime, turbulence intensity represented by the standard deviation of the vertical velocity $\sigma_w$ increased during daytime on 27 and 28 November even though the maximum downward solar radiation was relatively low on 28 November (Fig. 7h). Because the weak pressure system over Beijing, the diurnal variation of wind speed was not dramatic and the vertical variation of wind speed within several hundreds of meters above the surface was relatively small.

The $PM_{2.5}$ concentration at IAP went through significant oscillations before its final dramatic increase on 30 November

(Fig. 8a). Right after noon on 29 November, the $PM_{2.5}$ concentration decreased significantly until about 1800 LST (marked 1 in Fig. 8). After sunset, the $PM_{2.5}$ concentration increased to its temporal maximum value around 2200 LST (marked 2 in Fig. 8). Then it decreased sharply until the early morning of 30 November (marked 3 in Fig. 8). Around 0600 LST, the $PM_{2.5}$ concentration increased dramatically and reached its highest value of the day, 527 $\mu$g m$^{-3}$, around 2000 LST (marked 4 in Fig. 8). After that, the $PM_{2.5}$ concentration was relatively steady and varied $\pm$ 100 $\mu$g m$^{-3}$ until the night of 1 December when it

decreased to below 50 $\mu$g m$^{-3}$ (marked 5 in Fig. 8).

In the rest of this section, we analyze the evolution of the $PM_{2.5}$ concentration at IAP during the period between noon 29 November and the end of the severe pollution event. We investigate the total five stages marked in Fig. 8 one in each subsection from sections 4.1 to 4.5 and conclude the important pollutant transporting mechanisms in section 4.6.

## 4.1 Stage 1

On 29 November, the downward solar radiation measured at IAP was significantly larger than the previous cloudy day (Fig. 7a). The convective mixing increased in the morning as shown in both $\sigma_w$ and turbulence kinetic energy (TKE) (marked 1 in Figs. 8d and 8e). Normally convective turbulent mixing in the morning would vertically spread trace gases and aerosols accumulated in the nighttime SBL, leading to reduction of their concentrations near the surface during the daytime. However in this morning, the near-surface wind was from south (Fig. 8f), where air pollution was heavy. As a result of the pollutants transport from south, the concentrations of $PM_{2.5}$, $q$, and $CO_2$ increased slightly (Figs. 8a, 8b, and 8c) until the increase of the strong northwesterly wind arrived in the afternoon (Fig. 8g). The strong wind shear relative to the rough urban surface generated strong turbulent mixing as explained by Sun et al. (2017), and resulted in the strong upward transport of the polluted air and the downward transport of the less-polluted air from northwest, leading to the decreases of $PM_{2.5}$, $q$, and $CO_2$ (marked 1 in Figs.8a, 8b, and 8c). Thus, the shear-generated turbulent mixing contributed the reduction of the surface $PM_{2.5}$ concentration in the afternoon of 29 November.

## 4.2 Stage 2

Because of the strong heating during the daytime of 29 November, the air temperature was the highest during the 7-days period (Fig. 7b). When the downward solar radiation decreased in the afternoon, the reduced solar heating at the surface and the emission of the longwave radiation from the surface resulted in the surface cooling. Molecular thermal conduction at the cooling surface led to the air temperature decrease near the surface while the air above was still relatively warm from the daytime heating. Consequently the vertical air temperature difference became significant especially after the downward solar radiation approached zero after sunset (marked 2 in Fig. 8h), leading to the SBL (Fig. 8i). Associated with the strong stable stratification near the surface, the shear-generated turbulence near the surface was reduced significantly from its daytime value. The weak turbulent mixing under the very SBL constrained the vertical spreading of the surface trace gases and aerosols. Meanwhile, the wind direction near the surface started to be influenced by the mountains northwest of Beijing. During the development of the downslope wind, the wind within the tower layer changed gradually from easterly to northwesterly. As the wind direction changed to southerly, the polluted air was transported into Beijing. Because the nighttime SBL prevented the air with high aerosol, $q$ and $CO_2$ from being mixed up, the $PM_{2.5}$ concentration as well as $q$ and $CO_2$ increased sharply (marked 2 in Figs. 8a, 8b, and 8c).

## 4.3 Stage 3

Because the thermally induced terrain flow is due to the horizontal pressure gradient generated by the daytime heating and the nighttime cooling over the sloped terrain similar to land breezes (Sun et al., 1998), the relatively strong heating on 29 November would lead to the relatively strong downslope drainage flow. Under the influence of the relatively clean downslope flow, the $PM_{2.5}$ concentration in the vicinity of the mountains northwest of Beijing was reduced, and the high $PM_{2.5}$ concentration center was shifted to southeast of Beijing starting around 0100 LST on 30 November (Fig. 5). Because of the eastward spreading of the

downslope flow and the decreasing strength of the downslope flow with distance from the mountains, the $PM_{2.5}$ concentration at the most western station, MEN, started to decrease first in the early evening and lasted the longest until nearly noon on 30 November until the daytime convective mixing brought the heavy polluted air down (section 4.4). In contrast, the $PM_{2.5}$ concentration at southeast of Beijing, TON, which is far from the mountains, remained around 500 $\mu g\ m^{-3}$ throughout the night of 29 November (Figs. 2b and 9). As the downslope flow transported low-humidity and cold air from the relatively high elevation (Fig. 9), the relatively large humidity and the high $PM_{2.5}$ concentration at TON in the evening of 29 November further indicates that the downslope wind did not influence TON much. Because of the cold downslope flow, the SBL in the Beijing area was further strengthened. As a result of the horizontal transport of the less polluted air by the downslope flow, the surface $PM_{2.5}$ concentration reduced significantly at IAP.

Meanwhile, the wind direction within the IAP tower layer changed to northeasterly under the influence of the surface pressure system (marked 3 in Fig. 8f), which also brought in the less-polluted air as shown in the slight decreases of both surface $q$ and the $PM_{2.5}$ concentrations at TON where the downslope flow could not reach (Fig. 9). The relatively clean air from both the downslope flow and the northeasterly was also visible in the aerosol lidar images in the early morning hours on 30 November (Figs. 10a, 10b, and 10c). At the same time, the wind above about 500 m changed from northwesterly to southwesterly around the midnight of 29 November (Fig. 4). Because of the strong SBL, wind-shear generated turbulence eddies are confined to the relatively thin SBL below 150 m (Fig.8i) leading to decoupling between the rough urban surface and the air above the SBL. That is, the air flow above the SBL is not effectively reduced by the surface drag. As a result, the wind speed above the SBL is increased above the SBL, which is shown at 140 m and 280 m in the early morning of 30 November (marked 3 in Fig. 8g). The relatively strong southwesterly flow above the SBL effectively transported the polluted air from south to above the Beijing area (Fig. 10). The increased wind shear above the SBL enhanced turbulent mixing, leading to the downward transport of the polluted air to about 300 m around 0600 LST on 30 November, which is clearly observed with the relatively high backscatter density at FIN and the relatively high NRB at IUM and IAP from the aerosol lidars. Therefore, both the advection of the less polluted air into the Beijing urban area from the surrounding mountains and from northeast as a result of the synoptic system change as well as the small increase of turbulent mixing in the vertical spreading of the polluted air below the upper-level southwesterly flow contributed the significant decrease of the $PM_{2.5}$ concentration at IAP in the early morning of 30 November (Fig. 8a). The process also decreased the surface concentrations of $q$ and $CO_2$ (marked 3 in Figs. 8b and 8c).

## 4.4   Stage 4

In the morning of 30 November, the heavy polluted air above was efficiently transported by the southwesterly flow to over the Beijing area due to its decoupling from the rough surface, which resulted in the elevated polluted air above the surface SBL as shown in the lidar observations (Fig. 10). The elevated aerosol layer is consistent with the elevated humid air observed from the soundings at NGT because the polluted air from south is characterized with high humidity and high $PM_{2.5}$ concentration (Fig. 4). The absence of the elevated aerosol signal from the mini MPLs at IAP and IUM and from the ceilometer at FIN between 0700 and 1000 LST (Fig. 10) could be due to the fast decrease of the lidar power with distance (Davoust et al., 2014).

After the downward solar radiation increased in the morning of 30 November (marked 4 in Fig. 8a), the convective mixing increased as shown in both $\sigma_w$ and TKE (marked 4 in Figs. 8d and 8e). The enhanced convective mixing led to the coupling between the surface and the elevated polluted air resulting in downward transport of the heavy polluted air and the dramatic increase of the $PM_{2.5}$ concentration, air temperature, wind speed, $q$, and $CO_2$ within the CBL in the morning (marked 4 in Fig. 8). The convective mixing also enhanced the upward transport of the heavy polluted air south of Beijing; the upper southwesterly wind brought the heavy polluted air quickly above Beijing, leading to the enhanced elevated high-aerosol layer at IAP (Fig. 10a). Because of the surface $PM_{2.5}$ concentration near the mountains was reduced by the relatively clean downslope flow, the surface $PM_{2.5}$ concentration would increase with distance from Beijing southeastward. As the convective mixing transported the heavy polluted air into the convective boundary layer, it would take time for the upper southwesterly flow to transfer the heavy polluted air to over Beijing after the convective mixing started. The enhanced aerosol concentration in the elevated high aerosol layer above the IAP tower indeed increased with time (Fig. 10a). Because the growth of the CBL, this high aerosol layer appeared to be "lifted" gradually from 270 m to about 400 m between 0600 and 1200 LST on 30 November even though the entrainment of the polluted air into the CBL reduced the aerosol concentration right above the CBL (Fig. 10a). Because of the relatively shallow CBL at its early development, the aerosol concentration in the CBL below the elevated high aerosol layer increased dramatically. As the CBL continued to grow, the high aerosol air was completely engulfed into the CBL around 1100 LST, leading to the explosive increase of the $PM_{2.5}$ concentration. The role of the convective mixing on transporting the high polluted air down into the Beijing area is clearly demonstrated in the simultaneous sharp increases of the surface $PM_{2.5}$ concentration and the downward solar radiation in Fig. 8a because the downward solar radiation drives the CBL development. Due to the overwhelming coverage of concrete large surface elements in the urban area and the large surface roughness from urban canopies, the CBL developed faster over the urban area than the rural area, leading to the fast downward transport of the polluted air as shown at 0900 LST and 1100 LST in Fig. 5.

The convective mixing also transported the heavy polluted air downward east of Beijing, where the nighttime surface $PM_{2.5}$ concentration was relatively high and beyond the reach of the relatively clean downslope flow. As a result, the high $PM_{2.5}$ center oriented in the northeast-southwest direction was enhanced as well and extended northeastward during the daytime. Gradually the surface northeasterly wind contributed to the horizontal advection of $PM_{2.5}$ westward, leading to a relatively small enhancement of the $PM_{2.5}$ concentration in the afternoon as the $PM_{2.5}$ concentration became higher northeast of Beijing. The development of the SBL near the surface after the decrease of the downward solar radiation also contributed to the wind speed increase (Fig. 8g). However because the relatively strong convective mixing above the SBL was relatively strong, the wind speed change was relatively small. Overall, within eight hours from 0900 to 1700 on 30 November, the $PM_{2.5}$ concentration increased about 302 $\mu$g m$^{-3}$ at IAP. As a result of the reduction of the downward solar radiation by the high aerosol concentration with the possible early development of clouds on 30 November, the maximum downward solar radiation at 140 m on 30 November was only about a half of that on 29 November.

Based on previous studies in the region and around the world, the stagnant urban environment with increasing humidity and anthropogenic emissions of volatile organic compounds, nitrogen oxides, and sulfur dioxides provide favorable conditions for aerosol growth and secondary aerosol formation as a result of chemical reactions and hygroscopic growth of aerosols

(e.g., Mader et al., 1952; Van Dingenen et al., 2004; Putaud et al., 2004, 2010; Parrish et al., 2011; Guo et al., 2014; Zhang et al., 2015). Therefore, aerosol growth in the Beijing area might have also contributed the dramatic increase of the PM$_{2.5}$ concentration on 30 November as the high-humidity air was transported into the urban environment. Based on the numerical study of an air pollution event on January, 2013 by Wang et al. (2017), the emission from the Beijing urban area contributed more than 80% of the surface PM$_{2.5}$ concentration. Because the high-humidity air favorable for local aerosol growth was transported into the Beijing area and was associated with high aerosols, sorting out contributions of aerosols originated south of Beijing from local aerosol growth to the dramatic increase of aerosol concentration during the severe pollution event requires more observations and further investigations. Based on the detailed aerosol and turbulence observations presented above, at least a significant fraction, if it is not a major fraction, of the fast increase of the surface PM$_{2.5}$ concentration on 30 November can be explained by physical transferring processes associated with horizontal advection above the PBL and turbulent mixing within the PBL in the polluted area and the Beijing area. Higher surface PM$_{2.5}$ concentration south of Beijing was indeed observed on 30 November (Hao et al., 2017).

## 4.5 Stage 5

Around the midnight of 30 November, the air flow in the tower layer changed direction from northeasterly to northwesterly and the wind speed within the tower layer increased slightly under the influence of the mountain west and north of Beijing (Figs. 8f and 8g). The wind direction change brought in the less polluted air, resulting in the surface PM$_{2.5}$ concentration reduction of about 100 $\mu$g m$^{-3}$. Meanwhile, clouds moved in as indicated in the time series of the downward longwave radiation (not shown) and the MODIS image around noon (Fig. 6d). As a result of both clouds and extremely high aerosol concentration, the downward solar radiation at the surface was near zero even around noon on 1 December (Fig. 8a). Without surface heating and strong wind shear to generate turbulent mixing for transporting the heavy polluted air vertically and without effective horizontal advection to reduce the pollutant concentration, the heavy polluted air with the surface PM$_{2.5}$ concentration of about 500 $\mu$g m$^{-3}$ at IAP remained in the weakly stable urban boundary layer throughout 1 December until the significant increase of northwesterly wind arrived at midnight, which reduced the PM$_{2.5}$ concentration down to below 50 $\mu$g m$^{-3}$ within two hours after the wind speed increase.

## 4.6 Discussions

Overall, both turbulent mixing and advection are responsible for transporting the high-polluted air to the Beijing urban area and the PM$_{2.5}$ oscillations. Turbulent mixing is generated by either surface heating or wind shear and is to transport pollutants from high to low concentration areas. If the upper level concentration of a pollutant is low, vertical turbulent mixing would lead to the decrease of the surface concentration. If the vertical concentration gradient is reversed, vertical turbulent mixing leads to an increase of the surface concentration. Advection is to transport the air downstream. If the concentration of a pollutant is higher upstream than downstream, the downstream pollutant concentration would increase. In contrast to turbulent mixing, the downstream concentration would not affect the upstream concentration effectively. Horizontal advection of aerosols requires a non-zero air flow and an aerosol concentration difference between upstream and downstream. The air flow can be slowed

by the surface drag through the air-surface coupling by turbulent mixing. Strong turbulent mixing such as under convective or strong wind conditions leads to strong impacts of the rough surface on the air flow. When the air near the surface is stably stratified, the air flow is slowed down because turbulence eddies consume energy to transfer heat downward such that turbulent kinetic energy cannot increase significantly, which is commonly called the stable stability effect on air flow (e.g., Garratt, 1992). That is, the air flow is not fully coupled with the surface, therefore, the impact of the surface drag on the air flow is reduced. As a result, the air flow above an stable layer is not slowed down by the rough surface as much as under unstable or neutral conditions. Therefore, the stable boundary layer can not only trap pollutants as is commonly observed, but also assist the pollutant transport above. Consequently horizontal advection of aerosols can be much more effective over a stable boundary layer than through a convective boundary layer. During this severe pollution event, the effective horizontal transport of the heavy polluted air above the PBL and the convective mixing within the PBL led to the dramatic increase of the surface $PM_{2.5}$ concentration in the Beijing area. The roles of turbulent mixing and advection in the development of the severe air pollution event are schematically illustrated in Figure 11.

Elevated polluted air has been observed in the Beijing area (e.g., Han et al., 2018); advection process is commonly believed as the transporting mechanism. This study demonstrates that turbulent mixing in the PBL plays an important role in direct transporting polluted air and in determining effectiveness of horizontal advection of polluted air.

## 5   Summary

We analyzed the extreme heavy pollution event at the end of November 2015, Beijing, China based on the observations collected during SURF-15. Similar to previous pollution studies, we found that the slow movement of the regional pressure system and the influence of the mountains northwest of Beijing provided favorable conditions to keep polluted air in the Beijing area. In addition, we also identified the unique role of convective turbulent mixing and the stable boundary layer in pollutant transfer processes and their impacts on effectiveness of horizontal pollutant transport.

During the daytime of 29 November, the relatively strong downward solar radiation led to the significant warming of the boundary layer air, which provide a favorable condition for the development of the very stable nighttime boundary layer in the Beijing area. The downslope flow from the mountains northwest of Beijing brought in the relatively clean air, which reduced the surface $PM_{2.5}$ concentration in the northwest part of Beijing and the high $PM_{2.5}$ concentration appeared in the southeast part of Beijing. Around midnight on 29 November, the wind above the nighttime stable boundary layer changed from northwest-north to southwest. With the reduced surface drag through the reduced coupling of the stable boundary layer, the southwesterly wind effectively transported the polluted air from south of Beijing to above the urban area. After sunrise on 30 November, the daytime convective mixing with large coherent eddies transported the elevated polluted air downward, which initiated the dramatic increase of the surface $PM_{2.5}$ concentration in the morning of 30 November. Meanwhile, the vertical convective mixing kept transporting the heavy polluted air southwest of Beijing upward and the upper southwesterly wind kept transporting the polluted air to above the convective boundary layer over Beijing, which provided the continuous pollution source to be engulfed into the convective boundary layer leading to the explosive increase of the surface $PM_{2.5}$ concentration in Beijing. The cloud cover

as well as the high aerosol concentration on 1 December prevented buoyancy-generation of turbulence; weak winds prevented shear-generation of turbulence, leading to the stagnant high $PM_{2.5}$ air throughout 1 December until midnight when the strong northwesterly wind arrived, which reduced the $PM_{2.5}$ concentration to below 50 $\mu g\ m^{-3}$ in the Beijing area. The low wind and high humidity air during the build-up of the polluted air on 30 November might have also contribute local aerosol growth.

Significant oscillations of the surface $PM_{2.5}$ concentration transported from south of Beijing prior to the dramatic increase of $PM_{2.5}$ on 30 November can also be explained in terms of turbulent mixing and horizontal advection on pollutant transport.

Interactions between turbulent mixing and horizontal advection in transporting polluted air demonstrated in this study raise serious challenges on numerical models. Failure to simulate stable boundary layers in numerical models (e.g., Sun et al., 2015) may lead to unsuccessful prediction of aerosol concentrations not only because of the trapping mechanism of the stable

boundary layer but also because of its decoupling mechanism on horizontal transport of aerosols above. The roles of turbulent mixing and horizontal advection in transporting moisture and aerosols can provide useful information for chemical reactions as well.

*Data availability.* http://www.ium.cn:8088/dataCenter/

*Competing interests.* The authors declare that they have no conflict of interest

*Acknowledgements.* The authors would like to thank two anonymous reviewers for their helpful comments and to acknowledge the data collected by the IAP, Chinese Academy of Sciences. The SURF project is sponsored by Institute of Urban Meteorology, Meteorological Administration, Beijing, China, the Ministry of Science and Technology of China, Grant numbers 2015DFA20870 and 2016YFC0203302, the National Natural Science Foundation of China, Project No. 41505102, and Beijing Natural Science Foundation of China Grant No. 8171002. The University Corporation for Atmospheric Research manages the National Center for Atmospheric Research under sponsorship

by the National Science Foundation. Any opinions, findings and conclusions, or recommendations expressed in this publication are those of the authors and do not necessarily reflect the views of the National Science Foundation.

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

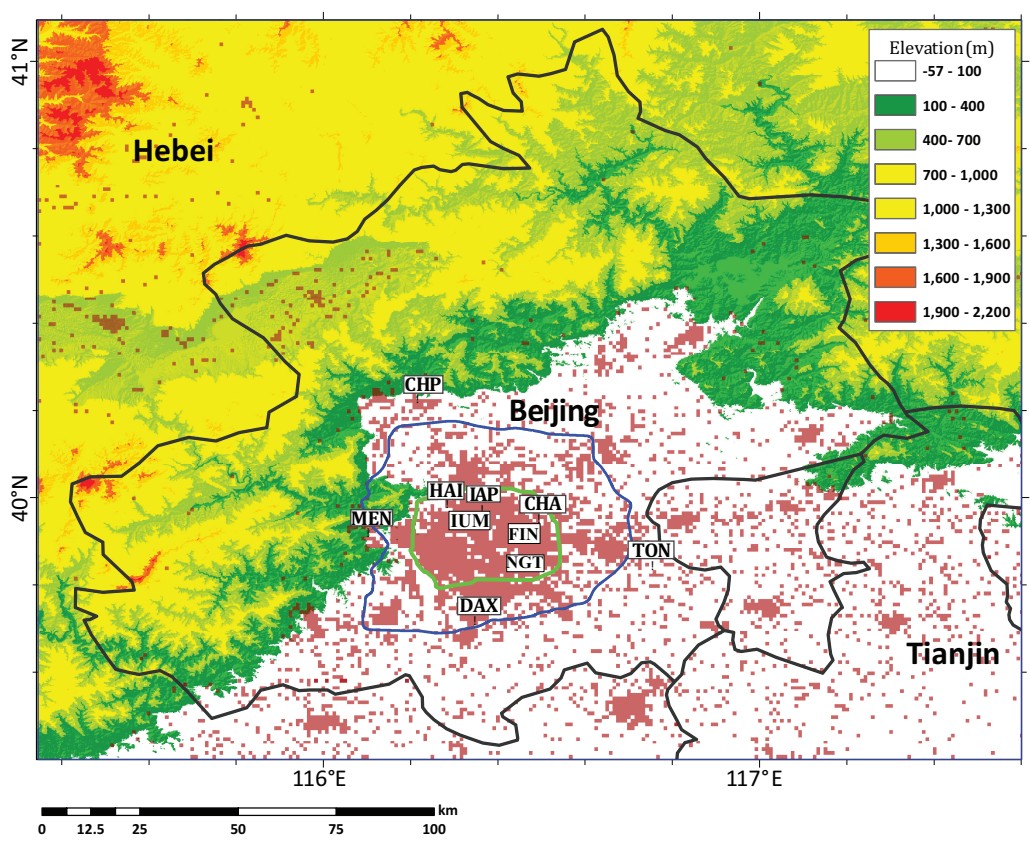

**Figure 1.** Map of topography around Beijing (bounded by the black curve with "Beijing" in its center), China, where the building areas (brown dots), the 6th (blue curve) and the 5th (green curve) ring roads, and the main observation sites used in this study are marked.

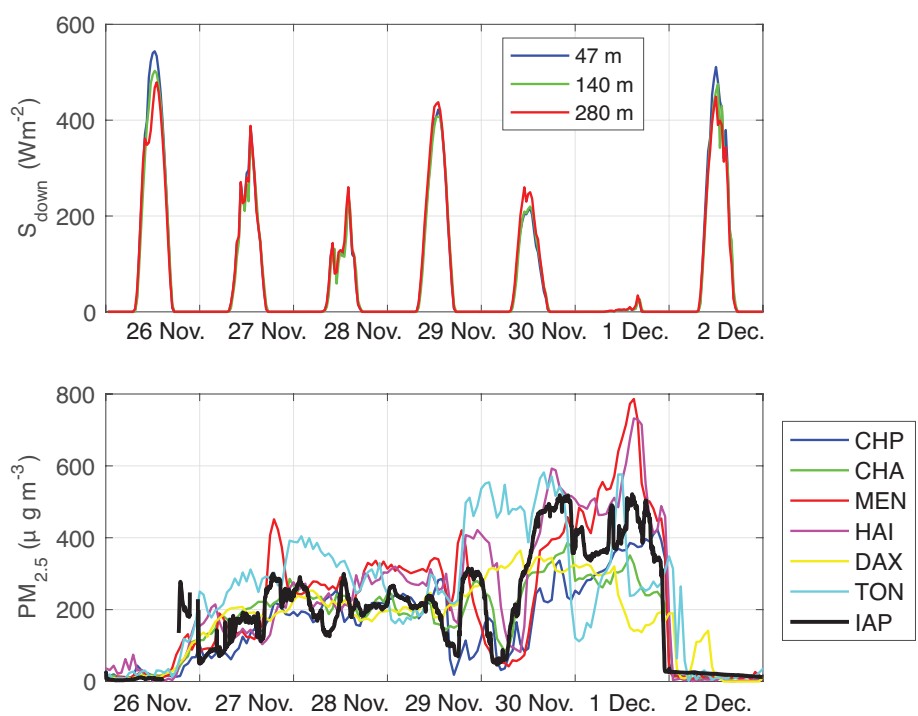

**Figure 2.** (a) The downward solar radiation $S_{down}$ at the labeled heights and (b) the temporal variation of the PM$_{2.5}$ concentration at the labeled stations.

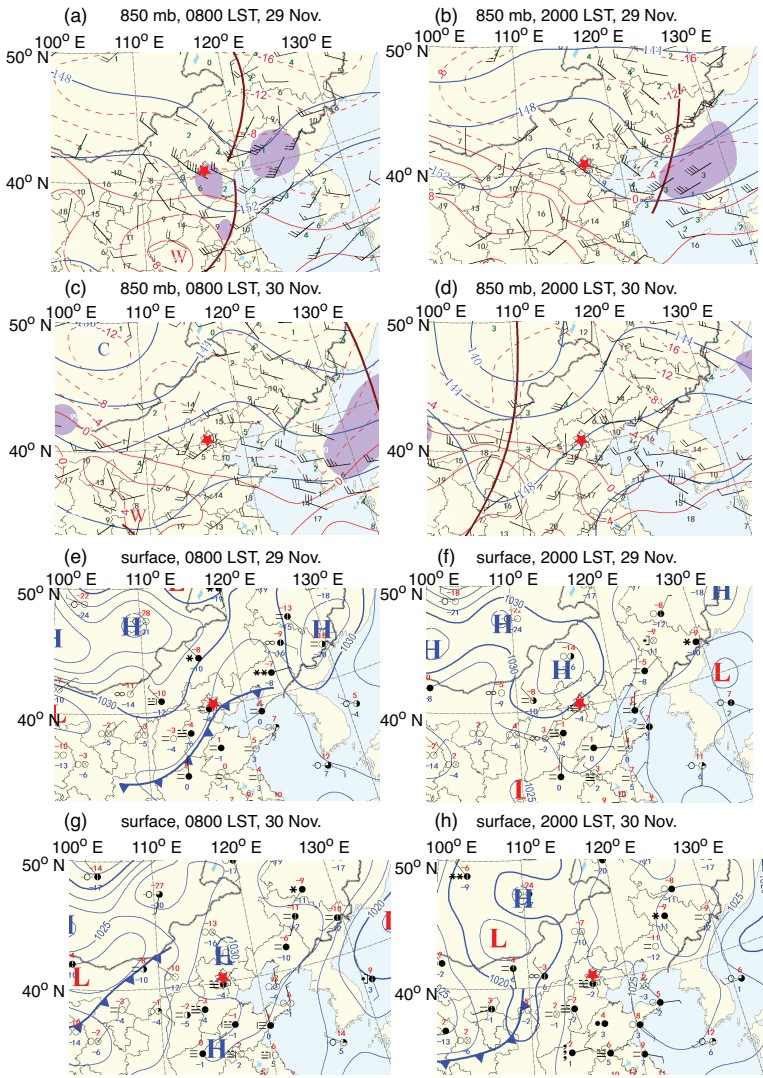

**Figure 3.** The 850-mb weather maps at (a) 0800 LST and (b) 2000 LST, and the surface maps at (e) 0800 LST and (f) 2000 LST on 29 November, and the 850-mb weather maps at (c) 0800 LST and (d) 2000 LST, and the surface maps at (g) 0800 LST and (h) 2000 LST on 30 November, where Beijing is marked by the red star in each panel.

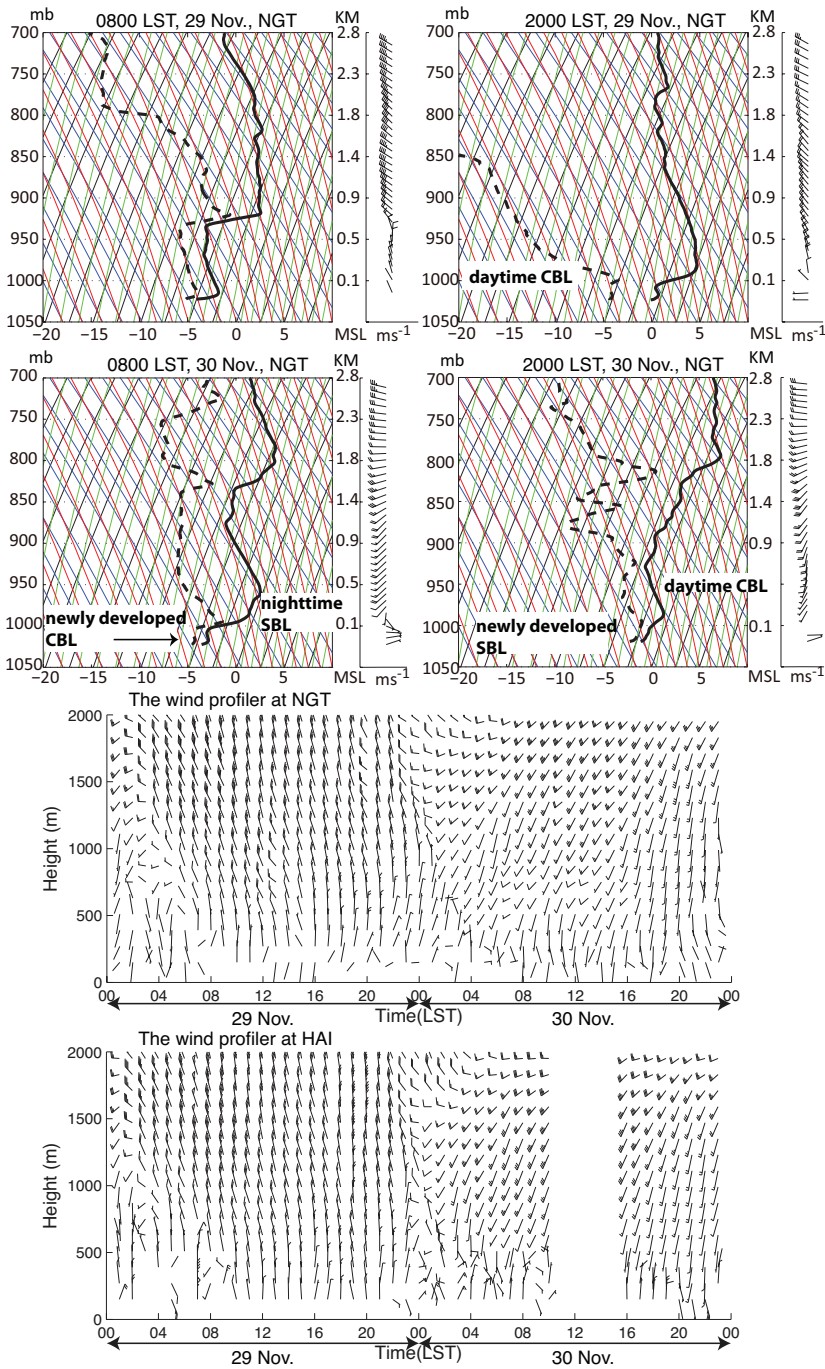

**Figure 4.** Temperature (solid), dew point (dashed), and wind vectors from the sounding profiles at 0800 LST and 2000 LST at NGT on 29 (top row) and 30 (the top second row) November, and the temporal variation of the hourly wind-vector profiles from the wind profiler at NGT (the top third row) and at HAI (bottom) for 29-30 November.

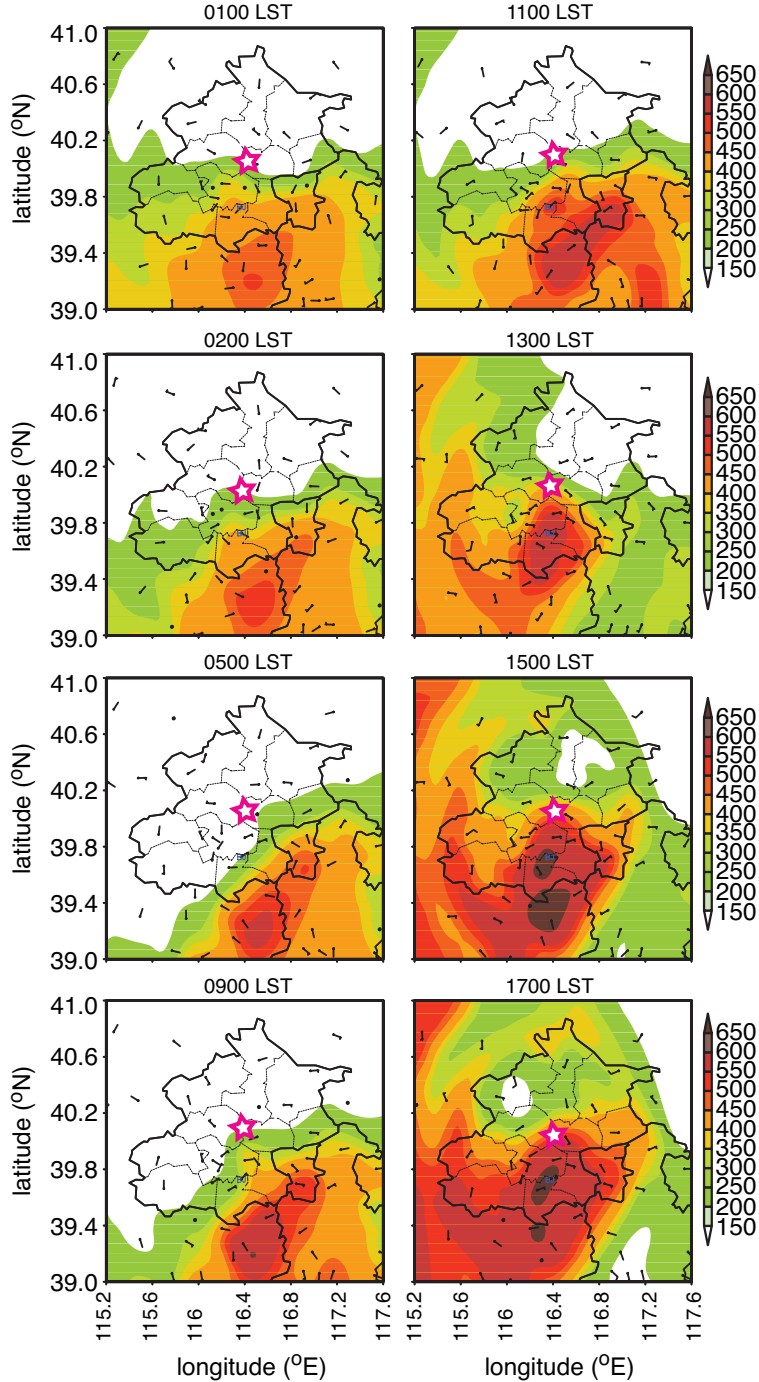

**Figure 5.** The surface PM$_{2.5}$ concentration in $\mu g\ m^{-3}$ at 0000 LST, 0200 LST, 0500 LST, 0900 LST, and 1100 LST, 1300 LST, 1500 LST, and 1700 LST on 30 November with surface wind barbs. The Beijing city boundary is marked by the dark black curve with the magenta star in its center.

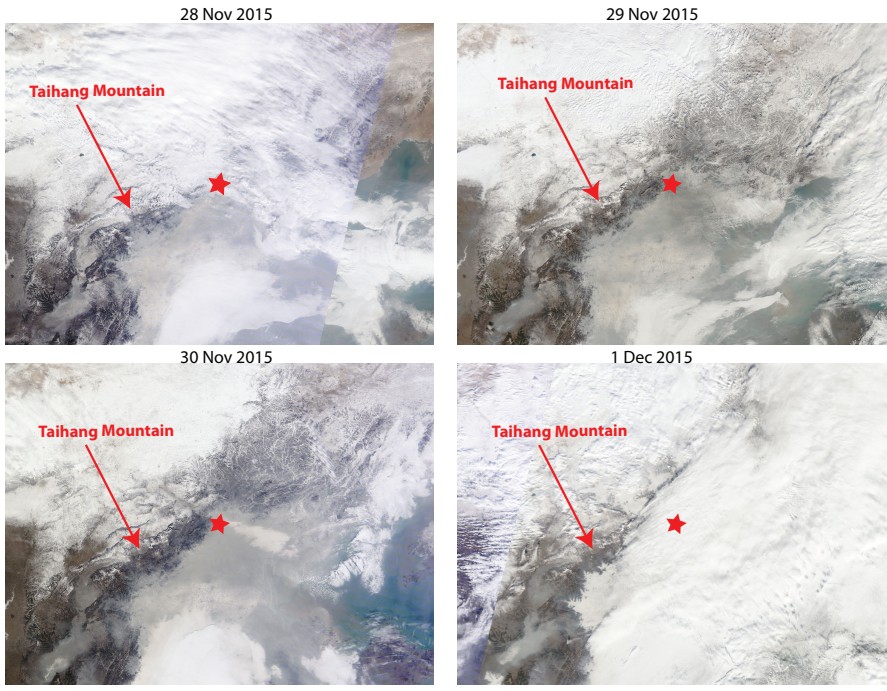

**Figure 6.** MODIS rapid response images around noon from 28 November to 1 December, 2015, where the red star marked the location of Beijing. The dark line southwest of the star visible on all the images is the Taihang Mountain Ridge.

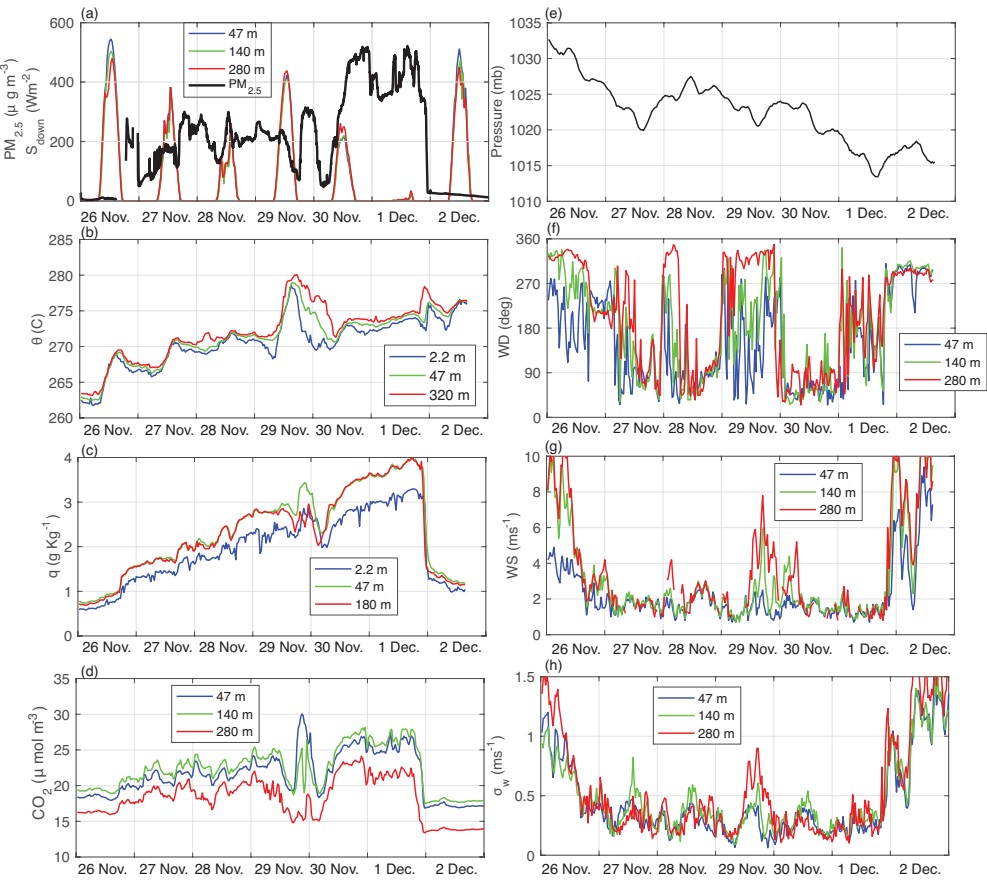

**Figure 7.** Temporal variations of (a) the $PM_{2.5}$ concentration and the downward solar radiation $S_{down}$, (b) the potential temperature $\theta$, (c) the specific humidity $q$, (d) the $CO_2$ concentration, (e) the pressure at 2.2 m, (f) wind direction $WD$, (g) wind speed $WS$, and (h) the standard deviation of the vertical velocity $\sigma_w$ at IAP from 26 November to 2 December.

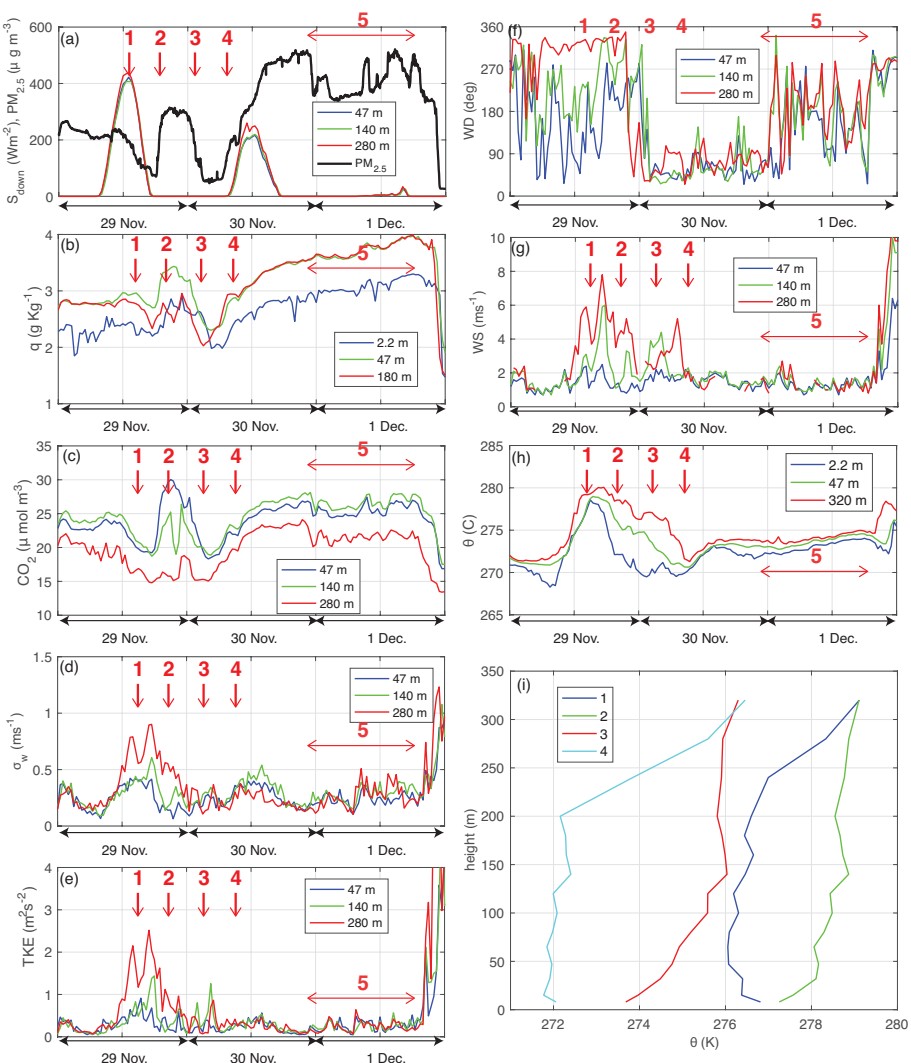

**Figure 8.** Temporal variations of (a) the 10-m $PM_{2.5}$ concentration and the downward solar radiation $S_{down}$, (b) the specific humidity $q$, (c) the $CO_2$ concentration, (d) the standard deviation of the vertical velocity $\sigma_w$, (e) turbulence kinetic energy (TKE), (f) wind direction $WD$, (g) wind speed $WS$, (h) the potential temperature $\theta$ at IAP from 29 November to 1 December, and (i) $\theta$ profiles from the IAP tower at the four times marked in (a)-(h).

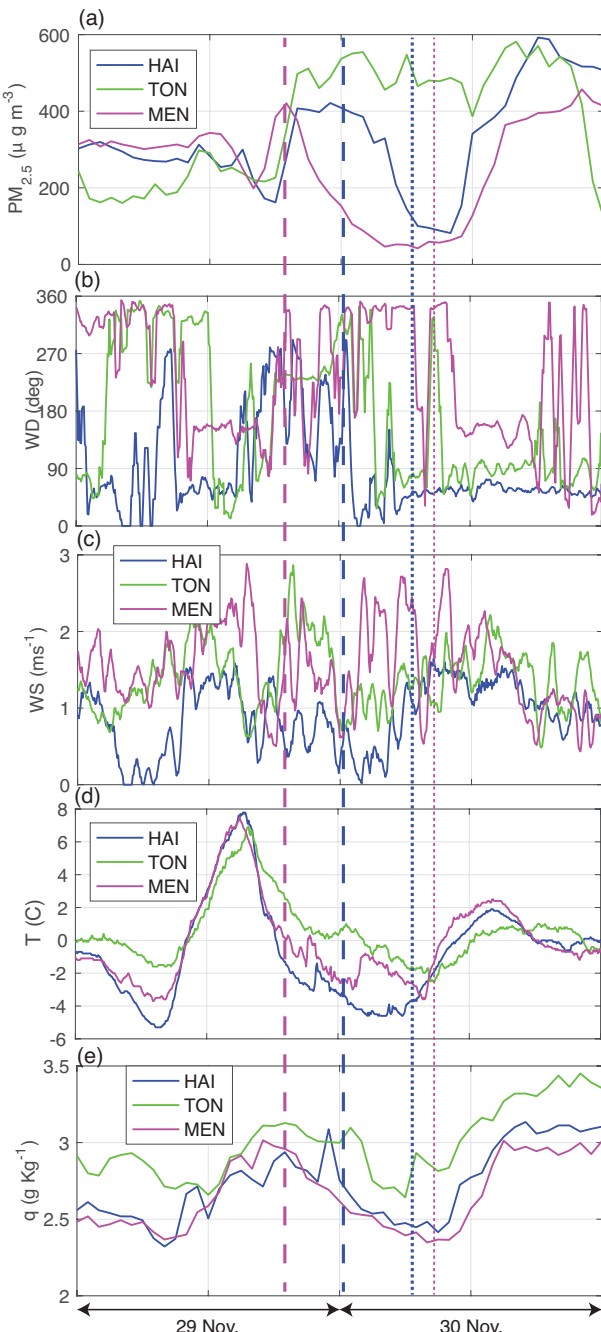

**Figure 9.** Temporal variations of (a) the surface $PM_{2.5}$ concentration, (b) wind direction $WD$, (c) wind speed ($WS$), (d) air temperature $T$, and (e) water vapor specific humidity $q$ at HAI, TON, and MEN from 29 to 30 November. The vertical dashed lines with the corresponding colors for MEN and HAI mark the $PM_{2.5}$ decreases (stage 3) due to the downslope flow and the wind direction change to northeasterly in the SBL, respectively. The vertical dotted lines mark the $PM_{2.5}$ increases (stage 4) associated with the development of the convective boundary layer at the two stations.

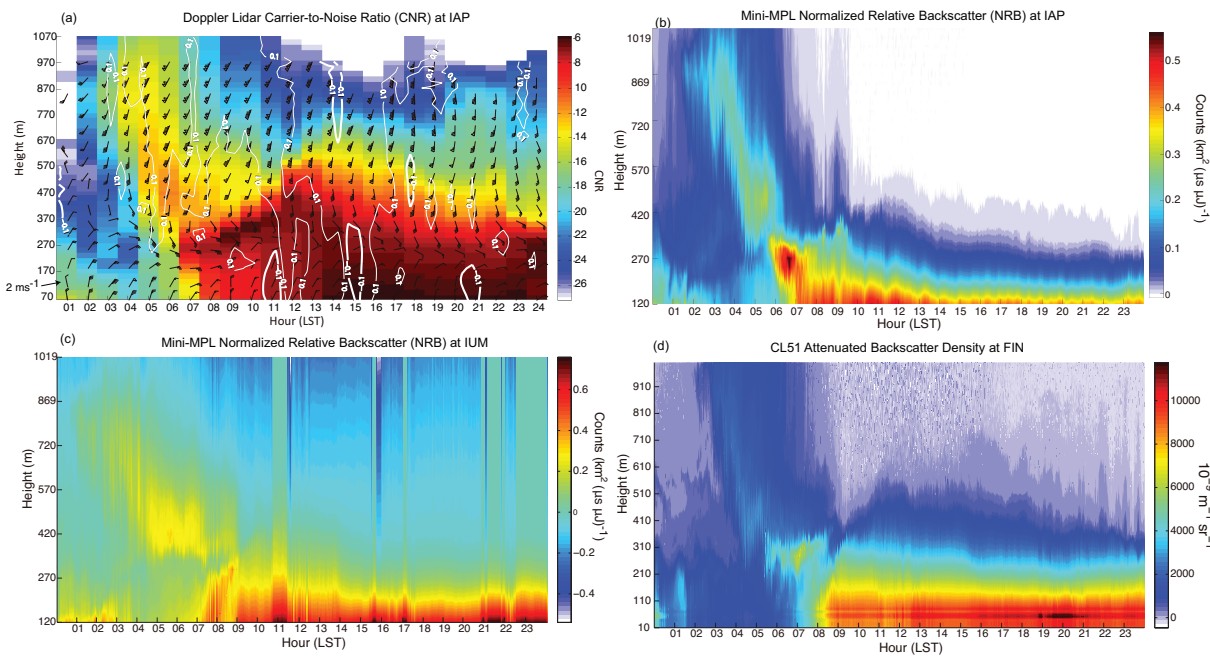

**Figure 10.** (a) The Doppler lidar carrier to noise ratio (CNR) with wind barbs at IAP, the normalized relative backscatter (NRB) from the mini-MPL lidar at (b) IAP and (c) IUM, and (d) the attenuated backscatter density from the CL51 lidar at FIN on 30 November. The wind barbs of 2 m s$^{-1}$ is marked in the bottom left corner of (a).

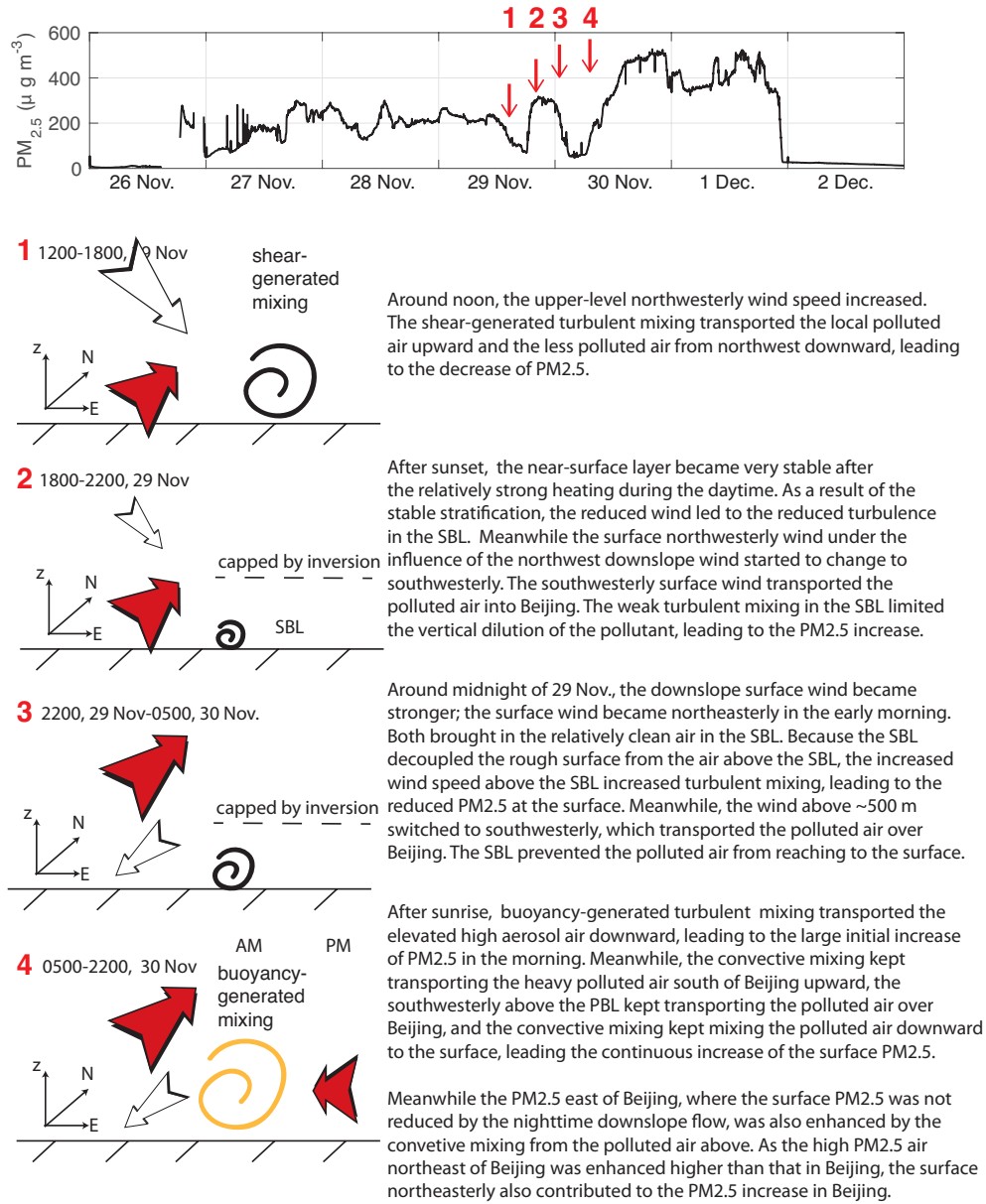

**Figure 11.** Schematic diagrams of the roles of turbulent mixing and advection in the four stages of the surface PM$_{2.5}$ oscillation (marked in the top panel) prior to and in the development of the heavy pollution event on 30 November at IAP. Thermally and mechanically forced turbulence eddies are represented by yellow and black spirals, respectively. The white and the red arrows represent the relatively clean and dirty air, respectively.