# Peer review of "Observational Analyses of Dramatic Developments of A Severe Air Pollution Event in the Beijing Area"

_Atmospheric Chemistry and Physics, 2017_

## Referee Comment (RC1) · Anonymous Referee #1 · 25 Nov 2017

This study investigates the meteorological reasons for haze formation in Beijing that occurred at the end of November 2015. The authors presented one-week surface stable layer and wind data collected at Beijing sampling site that of interest to the readers to solidify their findings. The paper is valuable to the audiences and scientists who want to understand the reasons for Beijing haze formations in winter and also well organized for being published on ACPD.

However, this study is unable to provide a long-term meteorological dataset of haze event at the sampling site in Beijing, where $\sim$30% of the year were haze day according to the Chinese air quality standards (Page 1 Line 22). Zheng et al collected one-year Beijing PM2.5 samples in 2013, when 73 haze episodes were found. They found these episodes differed in terms of sources and formation process and long term policies for

air quality regulation could not be determined by the results from the single episode. Please see Environment Science & Technology 2016, 50, 4632-4641.

In addition, the air pollution in Northern China exhibited a spatiotemporal variations, even in one city. Please see Environment Pollution 2017, 227, 334-347. In other words, the results from one sampling site in Beijing is unable to represent the whole one to illustrate the reasons of haze formation in Beijing.

Also, a large number of papers have already looked at this topic from the meteorological angle.

1. Characteristics of chemical composition and role of meteorological factors during heavy aerosol pollution episodes in northern Beijing area in autumn and winter of 2015 Author: Zhang, Zhouxiang; Zhang, Xiaoye; Zhang, Yangmei;et al. TELLUS SERIES B-CHEMICAL AND PHYSICAL METEOROLOGY, 69: 1347484, JUL 26 2017 2. Attributions of meteorological and emission factors to the 2015 winter severe haze pollution episodes in China's Jing-Jin-Ji area Author: Liu, Tingting; Gong, Sunling; He, Jianjun; et al ATMOSPHERIC CHEMISTRY AND PHYSICS, 17: 2971-2980 , FEB 27 2017 3. Characteristics and classification of PM2.5 pollution episodes in Beijing from 2013 to 2015 Author: Wang, Xiaoqi; Wei, Wei; Cheng, Shuiyuan; et al. SCIENCE OF THE TOTAL ENVIRONMENT , 612 : 170-179, JAN 15 2018 4. Relative Contributions of Boundary-Layer Meteorological Factors to the Explosive Growth of PM2.5 during the Red-Alert Heavy Pollution Episodes in Beijing in December 2016 Author: Zhong, Junting; Zhang, Xiaoye; Wang, Yaqiang; et al. JOURNAL OF METEOROLOGICAL RESEARCH, 31 : 809-819 , OCT 2017 5. Climatology of the Meteorological Factors Associated with Haze Events over Northern China and Their Potential Response to the Quasi-Biannual Oscillation Author: Liang, Ju; Tang, Yaoguo JOURNAL OF METEOROLOGICAL RESEARCH , 31 , 5: 852-864 , OCT 2017 6. Cause and predictability for the severe haze pollution in downtown Beijing in November-December 2015 Author: Zhang, Ziyin; Gong, Daoyi; Mao, Rui; et al. SCIENCE OF THE TOTAL ENVIRONMENT, 592 : 627-638 , AUG 15 2017 7. Characteristics of chemical composition

and role of meteorological factors during heavy aerosol pollution episodes in northern Beijing area in autumn and winter of 2015 Author: Zhang, Zhouxiang; Zhang, Xiaoye; Zhang, Yangmei; et al. TELLUS SERIES B-CHEMICAL AND PHYSICAL METEOROL-OGY, 69 : 1347484, JUL 26 2017 8. Local and regional contributions to fine particulate matter in Beijing during heavy haze episodes Author: Wang, Yangjun; Bao, Shengwei; Wang, Shuxiao; et al. SCIENCE OF THE TOTAL ENVIRONMENT, 580 : 283-296, FEB 15 2017

Thus, this paper could not show a more complete picture of illustrating the meteorological reasons for Beijing haze formation and lacks the comprehensiveness for the audiences in ACP . I recommend this paper could not be accepted by ACP.

---

## Author Comment (AC1) · 7 Dec 2017

We would like to thank Reviewer 1 for spending his/her valuable time on reviewing the paper. Obvious, we disagree with Reviewer 1 on values of detailed process studies vs. climatology studies. Hopefully through our detailed replies below (in red), Reviewer 1 can see the unique values of our contribution to understanding developments of pollution events at Beijing.

This study investigates the meteorological reasons for haze formation in Beijing that occurred at the end of November 2015. The authors presented one-week surface stable layer and wind data collected at Beijing sampling site that of interest to the readers to solidify their findings. The paper is valuable to the audiences and scientists

who want to understand the reasons for Beijing haze formations in winter and also well organized for being published on ACPD.

We have detailed data analyses on the severe air pollution event, which indeed lasted about a week. However, the boundary layer was not stable during the entire week-long time period and the dramatic increase of PM2.5 actually occurred during a day-time convective period. We clearly identified the role of the nighttime stable boundary layer in helping transferring air pollutants above Beijing and the dramatic increase of PM2.5 as a result of development of convective turbulent mixing in transporting pollutants down to Beijing in the morning. In other words, we found that the role of the stable boundary layer is beyond trapping pollutants within the stable boundary layer as frequently discussed in the literature. In addition, our observation coverage, which includes remotely sensed aerosol measurements at three locations across Beijing, vertical measurements of meteorological variables from remote sensing instruments and soundings, and turbulence measurements up to 280 m, is unprecedented in the area because of the special field campaign of SURF-15. Because the pollutant transfer mechanism described in the manuscript (not haze formation as the reviewer described) can occur at other places not just unique for the Beijing area, understanding this unique pollutant transfer mechanism can help improving pollution forecasts.

However, this study is unable to provide a long-term meteorological dataset of haze event at the sampling site in Beijing, where âĹij30

Climatological investigation of haze events is not the goal of this study. Detailed analyses with unprecedented observations in the Beijing area are lacking in the area in the literature, that is what we aim at.

In addition, the air pollution in Northern China exhibited a spatiotemporal variations, even in one city. Please see Environment Pollution 2017, 227, 334-347. In other words, the results from one sampling site in Beijing is unable to represent the whole one to illustrate the reasons of haze formation in Beijing.

[Figure]

Please check out section 2 on instrumentation and observations across Beijing. It is definitely not at one observation site.

Also, a large number of papers have already looked at this topic from the meteorological angle. 1. Characteristics of chemical composition and role of meteorological factors during heavy aerosol pollution episodes in northern Beijing area in autumn and winter of 2015 Author: Zhang, Zhouxiang; Zhang, Xiaoye; Zhang, Yangmei;et al. TELLUS SERIES B-CHEMICAL AND PHYSICAL METEOROLOGY, 69: 1347484, JUL 26 2017 2.

Their focuses are the composition of pollutants and the feedback effect of meteorological conditions after the formation of a pollution event. Our analyses focused on the meteorological condition that led to the rapid growth of PM2.5.

2. Attributions of meteorological and emission factors to the 2015 winter severe haze pol- lution episodes in China's Jing-Jin-Ji area Author: Liu, Tingting; Gong, Sunling; He, Jianjun; et al ATMOSPHERIC CHEMISTRY AND PHYSICS, 17: 2971-2980 , FEB 27 2017 3.

The observational analyses presented in the paper were mainly on correlations between the meteorological parameters such as air temperature and relative humidity at 2 m above the surface and wind speed at 10 m above the surface. Our analyses focused on what kind of physical transporting mechanisms were responsible for the dramatic increase of PM2.5, which is significantly different from this paper.

3. Characteristics and classification of PM2.5 pollution episodes in Beijing from 2013 to 2015 Author: Wang, Xiaoqi; Wei, Wei; Cheng, Shuiyuan; et al. SCIENCE OF THE TOTAL ENVIRONMENT , 612 : 170-179, JAN 15 2018 4.

The paper focused on model trajectory investigations of pollution episodes, while we used the unprecedented data coverage including in-site observations over a tall tower and remotely sensed measurements and found detailed horizontal and vertical transporting mechanisms for the dramatic increase of PM2.5 at Beijing.

4. Relative Contributions of Boundary-Layer Meteorological Factors to the Explosive Growth of PM2.5 during the Red-Alert Heavy Pollution Episodes in Beijing in December 2016 Author: Zhong, Junting; Zhang, Xiaoye; Wang, Yaqiang; et al. JOURNAL OF METEOROLOGICAL RESEARCH, 31 : 809-819 , OCT 2017 5.

5. Climatology of the Meteorological Factors Associated with Haze Events over Northern China and Their Potential Response to the Quasi-Biannual Oscillation Author: Liang, Ju; Tang, Yaoguo JOURNAL OF METE- OROLOGICAL RESEARCH , 31 , 5: 852-864 , OCT 2017 6.

As indicated from the title of the paper, it focused on climatology of meteorological variables with haze events, while we focused on detailed transporting mechanisms for our severe pollution event.

6. Cause and predictability for the severe haze pollution in downtown Beijing in November-December 2015 Au- thor: Zhang, Ziyin; Gong, Daoyi; Mao, Rui; et al. SCI-ENCE OF THE TOTAL ENVI- RONMENT, 592 : 627-638 , AUG 15 2017 7.

They explored possible influences of meteorological conditions on the severe pollution at Beijing. Again, we have different focuses.

7. Characteristics of chemical composition and role of meteorological factors during heavy aerosol pollution episodes in northern Beijing area in autumn and winter of 2015 Author: Zhang, Zhouxiang; Zhang, Xiaoye; Zhang, Yangmei; et al. TELLUS SERIES B-CHEMICAL AND PHYSICAL METEOROL- OGY, 69 : 1347484, JUL 26 2017 8.

This is the same paper as the first one.

8. Local and regional contributions to fine particulate matter in Beijing during heavy haze episodes Author: Wang, Yangjun; Bao, Shengwei; Wang, Shuxiao; et al. SCI-ENCE OF THE TOTAL ENVIRONMENT, 580 : 283-296, FEB 15 2017

This is a statistical study, which is very different from our focuses.

Thus, this paper could not show a more complete picture of illustrating the meteorological reasons for Beijing haze formation and lacks the comprehensiveness for the audiences in ACP . I recommend this paper could not be accepted by ACP.

We demonstrated how pollutants were transported horizontally and vertically to Beijing through detailed observational analyses across the Beijing area, which none of the suggested papers have shown. We focused on the dramatic development of a severe pollution event with the unprecedented vertical observations of aerosols at three locations across Beijing as well as turbulence data up to 280 m above the surface. Climatology studies are different from process analyses. Without process analyses to reveal detailed formation of some pollution events especially in vertical, we would not be able to understand what happens during the development of pollution events.

---

## Referee Comment (RC2) · Anonymous Referee #1 · 10 Dec 2017

Thanks for your responses.

I still hold on my comments. Our recent papers being submitted to Environmental Pollution all have four reviewers for each one. It is essential for publishing a high quality paper.

I recommend this paper should be sent to more reviewers for comments.

———————————————

---

## Referee Comment (RC3) · Anonymous Referee #2 · 13 Dec 2017

This manuscript present a detailed analysis of physical processes leading to a single air pollution episode observed in Beijing, China. The paper appears to be scientifically sound with no major errors. The paper is relatively well structured and clearly written.

The only major challenge with this paper is that, during the past couple of years, a large a number of analyses on air pollution episodes in Chinese cities have already been published. The authors should better bring up in this paper how it differs from or adds to this previous knowledge. Below is a list more detailed comments in this regard + a few other comment that should be considered when revising the paper.

In section 1, the authors summarize several factors found to contribute to the development of an air pollution episode. The authors should explicitly add here the existence of positive feedback processes (e.g. influence of high aerosol concentrations on radia-

tion and BL development discussed in a few recent papers) that enhance air pollutant concentrations further. The authors should also briefly summarize the remaining gaps in our understanding on how air pollution episodes form and evolve in these kind of enviroments.

At the end of section 1, the authors define the goal of this paper. In this context, it would be very important to specifically to list the scientific goals of the paper (i.e. which scientific questions this paper is aiming to answer) and how these goals are related to aims of this work.

Section 4 describes the evolution of the air pollution episode and is the most important original part of this paper. The subsections of this section should be numbered differently, i.e. 4.1.n, not 4.0.n. The second paragraph of section 4.0.5 does not fit there in its present form (except the last sentence referring to figure 11), but should be presented in a separate section. My suggestion is to add a new section (either as the final subsection into section 4 or as a separate short section 5) where the authors put their findings into a broader context, including i) how the pollution event described here compares with pollution events in general in the Beijing areas ii) how the results of this study add to our understanding on pollution event development, and iii) what broader-scale implications these results have.

Current Section 5 contains mainly a summary of the main findings, not real scientific conclusions. Such conclusions should be added here. In case the authors decide not to write a separate section 5 regarding the previous comment, but simply expand section 4, then some of that material could be presented in "Conclusions".

The caption of figure 11 should be improved to more clearly tie the upper panel of this figure to the 4 follwing figures.

---

## Author Response (AR1)

**Reply to Reviewer 1**

We would like to thank Reviewer 1 for spending his/her valuable time on reviewing the paper. Obvious, we disagree with Reviewer 1 on values of detailed process studies vs. climatology studies. Hopefully through our detailed replies, Reviewer 1 can see the unique values of our contribution to understand developments of pollution events at Beijing.

This study investigates the meteorological reasons for haze formation in Beijing that occurred at the end of November 2015. The authors presented one-week surface stable layer and wind data collected at Beijing sampling site that of interest to the readers to solidify their findings. The paper is valuable to the audiences and scientists who want to understand the reasons for Beijing haze formations in winter and also well organized for being published on ACPD.

We indeed had detailed data analyses on the severe air pollution event, which lasted about a week. However, the boundary layer was not stable during the week period. In addition, the pollutant transfer mechanism described in the manuscript (not haze formation as the reviewer claimed) is not just unique for the Beijing area. We clearly identified the role of the nighttime stable boundary layer in helping transferring air pollutants above it instead of trapping it within the stable boundary layer with detailed observations from various observation platforms and at various locations across Beijing. Different from the previous studies of air pollution in the Beijing area, we emphasized the role of turbulent mixing in the dramatic development of the air pollution event.

However, this study is unable to provide a long-term meteorological dataset of haze event at the sampling site in Beijing, where ~30% of the year were haze day according to the Chinese air quality standards (Page 1 Line 22). Zheng et al collected one-year Beijing PM2.5 samples in 2013, when 73 haze episodes were found. They found these episodes differed in terms of sources and formation process and long term policies for air quality regulation could not be determined by the results from the single episode. Please see Environment Science & Technology 2016, 50, 4632-4641.

Climatological investigation of haze events is not the goal of this study. Detailed analyses with unprecedented observations in the Beijing area are lacking in the area in the literature, that is what we aim at.

In addition, the air pollution in Northern China exhibited a spatiotemporal variations,

even in one city. Please see Environment Pollution 2017, 227, 334-347. In other words, the results from one sampling site in Beijing is unable to represent the whole one to illustrate the reasons of haze formation in Beijing.

Please check out section 2 on instrumentation and observations across Beijing. It is definitely not at one observation site.

Also, a large number of papers have already looked at this topic from the meteorological angle.

We discussed each publication recommended by Reviewer 1 below. These publications are related to air pollution in the Beijing area, but none of them have detailed turbulence transport analyses as we did. We referred these publications in the revised introduction section.

1. Characteristics of chemical composition and role of meteorological factors during heavy aerosol pollution episodes in northern Beijing area in autumn and winter of 2015 Author: Zhang, Zhouxiang; Zhang, Xiaoye; Zhang, Yangmei;et al. TELLUS SERIES B-CHEMICAL AND PHYSICAL METEOROLOGY, 69: 1347484, JUL 26 2017 2.

 Their focuses are the composition of pollutants and the feedback effect of meteorological conditions after the formation of a pollen event. Our analyses focused on the meteorological condition that led to the rapid growth of PM2.5.

2. Attributions of meteorological and emission factors to the 2015 winter severe haze pol- lution episodes in China's Jing-Jin-Ji area Author: Liu, Tingting; Gong, Sunling; He, Jianjun; et al ATMOSPHERIC CHEMISTRY AND PHYSICS, 17: 2971-2980 , FEB 27 2017 3.

The observational analyses presented in the paper were mainly on correlations between the meteorological parameters such as air temperature and relative humidity at 2 m above the surface and wind speed at 10 m above the surface. Our analyses focused on what kind of physical transporting mechanisms are responsible for the dramatic increase of PM2.5, which is significantly different from our paper.

3. Characteristics and classification of PM2.5 pollution episodes in Beijing from 2013 to 2015 Author: Wang, Xiaoqi; Wei, Wei; Cheng, Shuiyuan; et al. SCIENCE OF THE TOTAL ENVIRONMENT , 612 : 170-179, JAN 15 2018 4.

The paper focused on model trajectory investigations of pollution episodes, while we used unprecedented in-site observations over a tall tower and remote sensed measurements and found detailed horizontal and vertical transporting mechanisms for the dramatic increase of PM2.5 at Beijing.

4. Relative Contributions of Boundary-Layer Meteorological Factors to the Explosive Growth of PM2.5 during the Red-Alert Heavy Pollution Episodes in Beijing in December 2016 Author: Zhong, Junting; Zhang, Xiaoye; Wang, Yaqiang; et al. JOURNAL OF METEOROLOGICAL RESEARCH, 31 : 809-819 , OCT 2017 5.

The study also found that pollutants were transported from south of Beijing. However, due to lack of vertical observations, they used ECMWF reanalysis data to understand the vertical extend of pollutants during the high PM2.5 period. In contrast, we have vertical aerosol observations from remote sensed instruments as well as vertical observations of meteorological variables. We demonstrated temporal and spatial variations of aerosol and meteorological information for the formation of the severe pollution event.

Climatology of the Meteorological Factors Associated with Haze Events over Northern China and Their Potential Response to the Quasi-Biannual Oscillation Author: Liang, Ju; Tang, Yaoguo JOURNAL OF METEOROLOGICAL RESEARCH , 31 , 5: 852-864 , OCT 2017 6.

As indicated from the title of the paper, it focused on climatology of meteorological variables with haze events, while we focused on detailed transporting mechanisms for the severe pollution event.

5. Cause and predictability for the severe haze pollution in downtown Beijing in November-December 2015 Au- thor: Zhang, Ziyin; Gong, Daoyi; Mao, Rui; et al. SCIENCE OF THE TOTAL ENVIRONMENT, 592 : 627-638 , AUG 15 2017 7.

They explored possible influences of meteorological conditions on the severe pollution at Beijing. Again, we have different focuses.

6. Characteristics of chemical composition and role of meteorological factors during heavy aerosol pollution episodes in northern Beijing area in autumn and winter of 2015 Author: Zhang, Zhouxiang; Zhang, Xiaoye; Zhang, Yangmei; et al. TELLUS SERIES B-CHEMICAL AND PHYSICAL METEOROL- OGY, 69 : 1347484, JUL 26 2017 8.

This is the same paper as the first paper.

7. Local and regional contributions to fine particulate matter in Beijing during heavy haze episodes Author: Wang, Yangjun; Bao, Shengwei; Wang, Shuxiao; et al. SCIENCE OF THE TOTAL ENVIRONMENT, 580 : 283-296, FEB 15 2017

This is a statistical study, which is very different from our focuses.

Thus, this paper could not show a more complete picture of illustrating the meteoro- logical reasons for Beijing haze formation and lacks the comprehensiveness for the audiences in ACP . I recommend this paper could not be accepted by ACP.

We demonstrated how pollutants were transported to Beijing through detailed observational analyses across the Beijing area with special focuses on the role of turbulent mixing in the transporting mechanism, which none of the reference papers mentioned above have shown. We focused on the dramatic development of the pollution event with the unprecedented observations. Climatology studies are different from process analyses. Without process analyses to reveal detailed formation of some pollution events especially in vertical, we still cannot understand what happens 3-dimensionally during the development of pollution events. The uniqueness of our study is the role of turbulent mixing in transporting aerosols and contribution of convective and stable boundary turbulent mixing in horizontal advection of aerosols.

**Reply to Reviewer 2**

This manuscript present a detailed analysis of physical processes leading to a single air pollution episode observed in Beijing, China. The paper appears to be scientifically sound with no major errors. The paper is relatively well structured and clearly written.

We would like to thank Referee 2 for the helpful comments.

The only major challenge with this paper is that, during the past couple of years, a large a number of analyses on air pollution episodes in Chinese cities have already been published. The authors should better bring up in this paper how it differs from or adds to this previous knowledge. Below is a list more detailed comments in this regard + a few other comment that should be considered when revising the paper.

In the revised paper, we emphasized differences between the focus of this paper and those in the literature.

In section 1, the authors summarize several factors found to contribute to the development of an air pollution episode. The authors should explicitly add here the existence of positive feedback processes (e.g. influence of high aerosol concentrations on radiation and BL development discussed in a few recent papers) that enhance air pollutant concentrations further. The authors should also briefly summarize the remaining gaps in our understanding on how air pollution episodes form and evolve in these kind of enviroments.

Those are good points. In the revised paper, we highlighted the importance of turbulent mixing on air pollution transfer, which may differ from the traditional view. We emphasized the role of the stable boundary layer on air pollutant transfer, i.e., the advection process. We also pointed out the role of aerosol positive feedback on the stable boundary layer development at the final stage of the episode.

At the end of section 1, the authors define the goal of this paper. In this context, it would be very important to specifically to list the scientific goals of the paper (i.e. which scientific questions this paper is aiming to answer) and how these goals are related to aims of this work.

In the revised manuscript, we clarified our goals in the manuscript.

Section 4 describes the evolution of the air pollution episode and is the most important original part of this paper. The subsections of this section should be numbered differently, i.e. 4.1.n, not 4.0.n. The second paragraph of section 4.0.5 does not fit there in its present form (except the last sentence referring to figure 11), but should be presented in a separate section. My suggestion is to add a new section (either as the final subsection into section 4 or as a separate short section 5) where the authors put their findings into a broader context, including i) how the pollution event described here compares with pollution events in general in the Beijing areas ii) how the results of this study add to our understanding on pollution event development, and iii) what broader-scale implications these results have.

Thanks for catching on the format mistake. We changed the format in the revised manuscript.

The last stage of the episode may not be dramatic as the previous stages, but it is part of the event. We modified the text to describe the period after the oscillation as one of the focusing periods in the development of the pollution episode.  Now the new subsection 4.5 fits better in section 4.

Current Section 5 contains mainly a summary of the main findings, not real scientific conclusions. Such conclusions should be added here. In case the authors decide not to write a separate section 5 regarding the previous comment, but simply expand section 4, then some of that material could be presented in "Conclusions".

We prefer to explain the physical process in section 4, but to highlight the important scientific points in the last section. For unknown reasons, the ACP latex macro does not have a format for summary. In the revised manuscript, we made the last section as Summary; hopefully ACP will accept the format.

The caption of figure 11 should be improved to more clearly tie the upper panel of this figure to the 4 follwing figures.

We modified the figure caption for Fig. 11.

**The list of changes in the revised manuscript**

1) We modified every section, especially Introduction, section 4, and Summary (used to Conclusions) to emphasize what's new in the manuscript. The most important point of the manuscript is the role of turbulent mixing in polluted air transporting mechanisms: convective turbulent mixing can effectively transport aerosols, and weak turbulent mixing associated with the stable boundary layer can decouple the atmosphere in vertical such that the horizontal advection of aerosols can be enhanced.

2) We went through the literature and emphasized differences between our study and the previous studies in the area.

3) We followed the suggestion of Reviewer 2 and defined stage 5 for the time period between the time when the PM2.5 concentration reached to its peak value and the time when the high PM2.5 concentration disappeared.  We marked all the five stages in Fig. 8. By doing so, section 4 is better organized. In addition, we added section 4.6 to discuss the important physical processes of turbulent mixing during the event. The turbulent mixing processes have general turbulent mixing characteristics, and should be important in all pollutants transports.

4) In Summary, we shorten it to help readers to quickly get the key points of the paper that are significantly different from the previous studies. In addition, we added

challenging issues for future pollution transport studies.

5) We labeled the convective boundary layer (CBL) and the stable boundary layer (SBL) in the soundings in Fig. 4 to help readers to understand the development of turbulent mixing in the atmospheric boundary layer, which plays a key role in modulating the surface PM2.5 value.

6) We labeled Taihang Mountain clearly in Fig. 6.

7) We redid Fig.10b to fix the data misalignment issue.

8) We modified Fig. 11 to clearly highlight the role of turbulent mixing in transporting aerosols.

9) Because of the above mentioned numerous changes and we use latex, we don't provide a marked-up manuscript here.

---

## Author Response (AR2)

This paper provides important observational and analytical information in the understanding of the complex systems of a pollution event in Beijing. The materials worth to be published. I have the following minor comments/suggestions.

Thanks for helpful comments.

1. Introduction - 1st Paragraph, at the end: Cite sources of the numbers quoted. ("... heavy, and severe pollution days at 27.1%, 10.5%, 6.8% and 3.2% of the year, respectively")

Done.

2. Check the availability of the quoted website http://www.ium.cn:8088/dataCenter/; from my site it's not accessible (timed out).

Thanks for catching it. We changed the website address as http://www.ium.cn/dataCenter/

3. Section 4.1: The role of turbulence was explained; no mention of advection, one way or another.

We did mention the role of horizontal transport of pollutants but didn't mention the word "advection". We revised the sentence in the revised version to make it clear.

4. Section 4.2: "As the wind direction changed to southerly, the polluted air was transported into Beijing": Seems not relevant to this section and can be deleted.

This is the important part for explaining the PM2.5 increase in the first half of the night. The stable boundary layer only provides a "shelter" to cap whatever is near the surface. Because PM2.5 was low in the afternoon, the high PM2.5 concentration in the first half of the night has to be transported in horizontally. The observation indicates that during the relatively slow establishment of the northwesterly downslope flow, the wind direction rotated clockwise from easterly to northwesterly. As the wind direction was from south, the heavy polluted air was transported into Beijing and was trapped near the surface. We modified the sentence to make this explanation clear.

5. Not required for this paper publication: It would be great to create a visualization movie/animation of the events analyzed here!

We wish that we could have it. The schematics in Fig. 11 are our attempt to help readers to visualize what happened. Maybe the story will inspire someone to simulate the case and create a movie. One difficulty is to get the stable boundary layer right in numerical models.